# Detection of Periodontal Bone Loss on Periapical Radiographs—A Diagnostic Study Using Different Convolutional Neural Networks

**DOI:** 10.3390/jcm12227189

**Published:** 2023-11-20

**Authors:** Patrick Hoss, Ole Meyer, Uta Christine Wölfle, Annika Wülk, Theresa Meusburger, Leon Meier, Reinhard Hickel, Volker Gruhn, Marc Hesenius, Jan Kühnisch, Helena Dujic

**Affiliations:** 1Department of Conservative Dentistry and Periodontology, LMU University Hospital, LMU Munich, 80336 Munich, Germany; patrick.hoss@t-online.de (P.H.); uta.woelfle@med.uni-muenchen.de (U.C.W.); annika.wuelk@gmx.de (A.W.); theresa.meusburger@hotmail.com (T.M.); meierl.leon@gmail.com (L.M.); hickel@dent.med.uni-muenchen.de (R.H.); h.dujic@med.uni-muenchen.de (H.D.); 2Institute for Software Engineering, University of Duisburg-Essen, 45127 Essen, Germany; ole.meyer@uni-due.de (O.M.); volker.gruhn@paluno.uni-due.de (V.G.); marc.hesenius@uni-due.de (M.H.)

**Keywords:** artificial intelligence, bone loss, convolutional neural networks, deep learning, dental radiography, machine learning, periodontitis

## Abstract

Interest in machine learning models and convolutional neural networks (CNNs) for diagnostic purposes is steadily increasing in dentistry. Here, CNNs can potentially help in the classification of periodontal bone loss (PBL). In this study, the diagnostic performance of five CNNs in detecting PBL on periapical radiographs was analyzed. A set of anonymized periapical radiographs (*N* = 21,819) was evaluated by a group of trained and calibrated dentists and classified into radiographs without PBL or with mild, moderate, or severe PBL. Five CNNs were trained over five epochs. Statistically, diagnostic performance was analyzed using accuracy (ACC), sensitivity (SE), specificity (SP), and area under the receiver operating curve (AUC). Here, overall ACC ranged from 82.0% to 84.8%, SE 88.8–90.7%, SP 66.2–71.2%, and AUC 0.884–0.913, indicating similar diagnostic performance of the five CNNs. Furthermore, performance differences were evident in the individual sextant groups. Here, the highest values were found for the mandibular anterior teeth (ACC 94.9–96.0%) and the lowest values for the maxillary posterior teeth (78.0–80.7%). It can be concluded that automatic assessment of PBL seems to be possible, but that diagnostic accuracy varies depending on the location in the dentition. Future research is needed to improve performance for all tooth groups.

## 1. Introduction

Periodontitis is a prevalent dental health problem and can be classified as a major global challenge that affects developed and developing countries [1,2,3]. Triggered by bacterial colonization of the root surface, the host’s immune system reacts with inflammatory processes to the microbial transition from a symbiotic bacterial environment to that of dysbiotic pathogens, leading to loss of supporting tooth tissue, pocket formation, and ulceration of the pocket epithelium [4,5]. If the condition advances, periodontal bone loss (PBL) can occur as the principal pathological characteristic of periodontitis [6]. Moreover, severe periodontitis is a major cause of missing teeth in adults, leading to reduced oral functioning and ultimately having an adverse effect on general health [7,8]. In this context, the link between periodontal disease and various systemic diseases such as cardiovascular diseases [9], diabetes [10], and respiratory diseases [11] should be emphasized. Considering the mostly irreversible consequences of periodontal disease, frequent periodontal screening is essential for the treatment of all patients and should be part of routine oral inspection [12]. According to the new guidelines introduced by the workshop on the classification of periodontal and peri-implant diseases and conditions [13,14], the evaluation of clinical attachment loss as well as the radiographic assessment of PBL has become critical in categorizing periodontitis into specific stages and subsequently in indicating optimal disease management. Nevertheless, both the clinical measurements and the radiographic assessment of PBL remain controversial in terms of their reliability. The measurement of clinical attachment loss by periodontal probing varies due to individual probing force, probe angulation, and varying probe tip diameter [15,16]. In addition, radiographic PBL evaluation represents a challenging task for a clinician due to possible variations in contrast and exposure angle as well as structural overlap, so that the interpretation of dental radiographs may lead to inconsistencies among dentists [17,18,19]. Here, the use of artificial intelligence (AI)-based diagnostics could reduce these diagnostic discrepancies. Consequently, several work groups have investigated the use of AI-based methods for automatized PBL detection on periapical radiographs [19,20,21,22,23,24,25,26,27,28,29] and panoramic X-rays [18,30,31,32,33,34,35,36,37,38,39,40]. In these studies, on the one hand, convolutional neural networks (CNNs) have shown potential in accurately detecting PBL on radiographs. However, due to differing CNNs and varying data sets, the existing studies show significant heterogeneity and, therefore, are difficult to compare [41,42,43]. In addition, little is known about whether different CNNs or anatomical regions influence diagnostic performance. Therefore, the aim of this study was to evaluate the diagnostic performance of five commonly used CNNs for automated PBL detection on periapical radiographs representing all sextants (upper and lower posterior teeth and upper and lower anterior teeth) and to statistically report their diagnostic performance with standardized variables, avoiding non-comparable results. In detail, it was first hypothesized that the diagnostic performance of the tested CNNs would have an accuracy of at least 90%. Secondly, diagnostic accuracy was hypothesized to be the same between all CNNs and anatomical regions.

## 2. Materials and Methods

### 2.1. Study Design

The Ethics Committee of the Medical Faculty of the Ludwig-Maximilians University of Munich approved this study protocol with project number 020-798. The recommendations of the Standard for Reporting of Diagnostic Accuracy Studies (STARD) steering committee [44] and the recommendations for the reporting of AI studies in dentistry [45] were followed in the study report.

### 2.2. Periapical Radiographs

For this study, anonymized periapical radiographs taken at the Department of Conservative Dentistry and Periodontology (Dental School of the LMU) and other dental practices were used. A high-quality image sample was secured by excluding inadequate X-rays, e.g., distorted images, images with incomplete teeth, or radiographs with implants. Following these exclusion criteria, a data set with 21,819 periapical radiographs stored in jpg format was assembled.

### 2.3. Categorization of Periodontal Bone Loss (Reference Standard)

Prior to the start of the study, a two-day workshop was held by the principal investigator (J.K.), during which the group of participating dentists (*N* = 7) was trained. In addition, the efficiency of the training was determined during a calibration course. Reproducibility of PBL within and between investigators was assessed using 150 periapical radiographs, and the corresponding inter- and intra-examiner reliability showed substantial kappa values [17]. The detailed kappa values are specified in Table 1. A group of graduated dentists (P.H., T.M., A.W., L.M.) then pre-categorized all X-rays by differentiating between healthy periodontium and mild, moderate, or severe PBL [13,14]. Following this, more clinically experienced examiners (H.D., U.W., J.K.) independently counterchecked each diagnostic decision. More specifically, these diagnostic criteria and ratings were applied: 0—healthy periodontium, PBL not detectable, 1—mild radiographic PBL up to 15% in the coronal third of the tooth, 2—moderate radiographic PBL between 15% and 33% of the root length, and 3—severe radiographic PBL beyond the coronal third of the tooth (Figure 1). In case of differing diagnostic opinions, each image was subject to continued discussion until consensus was achieved. The use of anonymized periapical radiographs meant that no further clinical information could have been acquired to make a diagnostic decision. One dichotomized diagnosis decision (0 vs. 1–3) was made for each X-ray, which consequently became the reference standard for the cyclic training and the repeated evaluation of the AI-based CNN.

### 2.4. Training of the Deep-Learning-Based CNNs (Test Method)

Hereafter, the utilized pipeline of well-established methods for developing the AI-based algorithm is explained. Initially, the whole image set of 21,819 periapical radiographs was subdivided into a training set (*N* = 18,819) and a test set (*N* = 3000). The latter was randomly selected from the entire data set, ensuring that all sextants were equally represented. This served as an independent test set for evaluation purposes only and was not included in the model training.

By using Python (version 3.8.5, https://www.python.org accessed on 17 November 2023) in conjunction with the PyTorch library (version 1.12.0, https://pytorch.org accessed on 17 November 2023), the training set was augmented so that the variability of the included radiographs could be improved. Therefore, images were modified using different transformations: random rotation up to 180 degrees, random changes in brightness, contrast, and saturation up to 20% with color jitter, and random affine transformation (translation up to 30% of the image size and zooming out up to 70%). As a result, a new, unique, and virtual grayscale image (RGB format) was created.

The augmented images were used to train the following pretrained CNNs: ResNet-18 [46], MobileNet V2 [47], ConvNeXT/small, ConvNeXT/base, and ConvNeXT/large [48]. The batch size amounted to 16 randomly selected images. The random selection of the respective images into batches was done using PyTorch’s built-in DataLoader class. The learning performance was repeatedly verified with the test set after 30 training steps. All CNNs were trained using backpropagation to determine the gradient for learning. Furthermore, the training was accelerated using Floating Point 16 and a university-based computer (i9 10850K 10 × 3.60 GHz, Intel Corp., Santa Clara, CA, USA) equipped with 48 GB RAM and a professional graphic card (GeForce RTX 3060, Nvidia, Santa Clara, CA, USA). Each CNN was trained over 5 epochs, with cross entropy loss as an error function and an application of the Adam optimizer (Betas 0.9 and 0.999, Epsilon × 10^−8^).

### 2.5. Statistical Analysis

The data were analyzed using Python (version 3.8.5). By computing the number of true positives (TPs), false positives (FPs), true negatives (TNs) and false negatives (FNs), the diagnostic accuracy (ACC = (TN + TP)/(TN + TP + FN + FP)) was identified. The sensitivity (SE), specificity (SP), positive predictive values (PPVs), negative predictive values (NPVs), and the area under the receiver operating characteristic (ROC) curve (AUC) were calculated with respect to the utilized CNN [49].

## 3. Results

For the purpose of this study, a total of 21,819 periapical radiographs were selected and divided into sextants (upper and lower posterior teeth as well as upper and lower anterior teeth). The image distribution in relation to the anatomical region and the PBL can be taken from Table 2. While the number of radiographs from the upper jaw was found to be comparable to that from the lower jaw, the overwhelming majority of images originated from posterior teeth compared to anterior teeth. Moreover, most included periapical radiographs showing teeth affected by mild PBL (42.6%). In contrast, radiographs with severe PBL had a notably lower proportion (6.9%) in the total data set.

The overall diagnostic performance for automatized detection of PBL on periapical radiographs in relation to the CNNs used are specified in Table 3 and Table 4. The CNNs achieved an overall ACC between 82.0% and 84.8%. The associated AUC values ranged from 0.884 to 0.913. Moreover, all tested CNNs showed consistently higher SE values varying between 88.8% and 90.7% compared to the SP values, which ranged from 66.2% to 71.2%.

When investigating the diagnostic performance of the CNNs depending on the anatomical region (Table 5 and Table 6), better results were mainly documented for mandibular teeth compared to maxillary teeth. In the anterior region, ACC values from 94.9% to 96.0% were observed for mandibular teeth and from 86.0% to 88.6% for maxillary teeth. When considering posterior teeth only, the ACC ranged from 82.2% to 86.1% for mandibular teeth and varied between 78.0% and 80.7% for maxillary teeth. In principle, the same tendency was also observed for the AUC values (Table 6).

All five CNNs, ResNet-18 (ACC 82.8%; AUC 0.884), MobileNetV2 (82.0%; 0.884), ConvNeXT/s (83.9%; 0.903), ConvNeXT/b (84.8%; 0.911) and ConvNeXT/l (84.2%; 0.913), tended to show similar performance data (Table 4). Furthermore, the hierarchy of results is evident in the receiver operating characteristic (ROC) curves of the five CNNs used to graphically compare diagnostic performance in detecting PBL (Figure 2).

## 4. Discussion

The present study was able to demonstrate that different CNN architectures are able to detect PBL on periapical radiographs. However, with an overall accuracy between 82.0% and 84.8%, none of the CNNs tested were able to achieve the primary expected accuracy of 90%. Although the CNNs achieved similar diagnostic performance compared to one another, there were differences for the various sextants. This led to the rejection of the originally formulated hypothesis. Nevertheless, the results obtained provide important information for the discussion.

When considering the ability of the tested CNNs to detect PBL in relation to sextants on periapical radiographs, differences between teeth in the lower and upper jaw were observed (Table 6). Here, the projection technique and overlaying anatomical structures such as the maxillary sinuses or the nasal cavities may have negatively affected the diagnostic performance in the upper jaw. In contrast to the maxilla, mandibular sextants can be captured more accurately by use of the right-angle technique, which results in less distorted images and better diagnostic performance data (Table 6). The previously mentioned factors most likely explain the documented differences in the model performance among sextants, which were found to be similar throughout all included CNNs (Table 6). Such differences are of methodological importance. For example, Tsoromokos et al. [24] included only periapical radiographs with mandibular teeth in their pilot study to avoid data inconsistencies. Additionally, other author groups excluded radiographs from some sextants [20] or vertically rotated maxillary to mandibular teeth [26]. Such procedures may have resulted in biased and/or noncomparable results. Consequently, aiming at increasing the comparability of future studies, it is suggested to provide data for each sextant based on a well-powered image sample.

The diagnostic performance between the included CNNs was found to be similar. In general, our study results are basically in line with recently published studies of similar methodologies for evaluating PBL on periapical radiographs [19,20,21,22,23,24,26,29]. For example, Lee et al. [26] presented a model that could detect periodontally compromised premolars and molars with a diagnostic accuracy of 82.8% and 73.5%, respectively. As part of the PBL assessment, Chen et al. [25] compared so-called fast and faster R-CNNs and then determined the severity of PBL. Unfortunately, no detailed accuracy values were provided [25]. Lee et al. [23] trained a machine learning model with precisely annotated periapical radiographs, which also classified PBL according to the latest classification [13]. In this context, high AUC values of 0.89, 0.90, and 0.90 were obtained for stages I, II, and III, respectively [23]. Another study with an accurate annotation process was introduced by Chen et al. [29]. Here, the model based on deep CNN algorithms provided an accuracy of 97% for the detection of PBL on periapical radiographs and showed superior performance compared to dentists. To the best of our knowledge, no study has compared multiple CNNs for PBL detection on periapical radiographs. In the literature, there is only one similarly designed study available that tested different CNNs to identify implant characteristics on periapical radiographs [50]. When also considering studies that analyzed panoramic X-rays for the presence of PBL, it can be concluded that the model metrics were found to be similar [18,30,31,32,33,34,36,37,38,39,40]. For instance, Krois et al. [38] presented a deep feed-forward CNN to detect PBL on image segments from panoramic radiographs. They chose binary decision making to distinguish between the presence or absence of PBL by introducing a cut-off value (20%, 25%, and 30%). A mean accuracy of 81% for PBL detection was achieved by the utilized CNN. In addition, the panoramic radiographs were manually cropped, focusing on a single tooth, and the images were flipped vertically by 180 degrees during pre-processing. Subsequently, it can be seen from the results that the diagnostic performance was validated in certain subgroups of teeth, with the highest accuracy value being reported for molars (86%). The deep learning model proposed in the study of Jiang et al. [30] was also applied to detect PBL on panoramic radiographs. The diagnostic performance of the model varied between 71% and 81% for different tooth groups. Interestingly, lower accuracy values were obtained not only for maxillary molars but also for mandibular anterior teeth, suggesting that overlapping anatomical structures may negatively impact the diagnostic performance for the anterior region in panoramic radiographs. Furthermore, the diagnostic performance for each periodontal stage was compared between the model and dentists. At all stages, the model achieved higher accuracy and sensitivity values compared to the dentists. Considering the reported results, it is worth noting that the author groups that accurately annotated PBL or features of PBL on panoramic radiographs generally published more favorable results [29,32,34,37].

This study has strengths and limitations. In view of the significant heterogeneity that previous studies have shown not only in their data sets (e.g., excluding certain tooth groups, the number of radiographs) but also in the evaluation method of diagnostic performance, then the training of commonly used CNNs with a data set representative of all sextants and the representation of their diagnostic performance with standardized variables can be considered a strength of this study [24,41,42,43]. Establishing a representative image data set for a particular finding with a relevant number of images can be considered a crucial factor. When comparing studies in terms of the total number of periapical radiographs, our study revealed a large data set (*N* = 21,819). Only Kearney et al. [51] utilized a larger data set, with over 100,000 radiographs; however, this study differed from our study methodologically by determining the clinical attachment level instead of PBL. Additionally, studies with panoramic radiographs should be mentioned in this context. With the exception of Kim et al. with more than 12,000 radiographs [37], almost all identified studies reported data sets with less than 2000 panoramic radiographs [18,30,31,32,33,34,35,36,38,39,40]. Moreover, our study allows the comparison of different CNNs for detecting PBL for each sextant. In addition, the data set included periapical radiographs with a broad spectrum of dental pathologies or restorations. 

As a limiting factor of our study, the unbalanced image distribution across all sextants should be discussed. Although the number of radiographs from the maxilla was found to be similar to that of the mandible, less than half of the images were available from anterior teeth compared to posterior teeth (Table 2), which possibly indicates an imbalance in the data set. The main reason leading to this unequal image distribution might be that under clinical conditions, the justification of an indication for radiography varies between the different sextants. In addition, moderate and severe PBL were also underrepresented. Such imbalances may negatively influence the diagnostic performance of CNNs. Therefore, it is crucial to safeguard a representative and well-balanced number of images for each sextant and severity score in order to improve the metrics of the models. Furthermore, this study utilized periapical radiographs only. However, both panoramic and periapical radiographs are considered relevant for PBL assessment. As for the aspect of comparing the diagnostic performance within different sextants, panoramic radiographs might be considered less applicable, since overlapping anatomical structures could potentially limit the diagnostic performance for the anterior region. Moreover, our data set was compiled from anonymized periapical X-rays; thus, no conclusions can be drawn about further, patient-specific diagnostic information. Additional diagnostic information, such as clinical attachment loss and pocket depths, would be particularly helpful for the initial diagnosis of periodontal disease, considering that the radiographic assessment of periodontal bone defects of low depth and buccolingual width might be restricted [52]. Here, the radiographic assessment of PBL becomes more relevant with further disease progression when the extent of osseous lesions can be visualized more accurately [53]. Another limitation to be mentioned is that we made a diagnosis for each image by distinguishing between a healthy periodontium and teeth affected with PBL (score 0 vs. 1–3). Considering that none of the five CNNs showed the hypothesized accuracy of 90%, this binary decision-making has to be understood as a limitation, which also negatively influenced the metrics of the models. It can be assumed that the precise annotation of PBL-related structures may increase the performance of the CNNs [23,29]. However, exact image labelling is time-consuming and requires extensive resources, especially with such large data sets. Nevertheless, it can be expected that precisely annotated radiographs representing a large and balanced data set would probably increase the precision of machine-based PBL detection.

## 5. Conclusions

In summary, the CNNs used showed nearly identical diagnostic performance in detecting PBL on periapical radiographs. However, different outcomes were documented among sextants, which can be primarily explained by the radiographic anatomy. With regard to comparable projects in the future, it is expected that the diagnostic performance can be further increased by precise annotations.

## Figures and Tables

**Figure 1 jcm-12-07189-f001:**
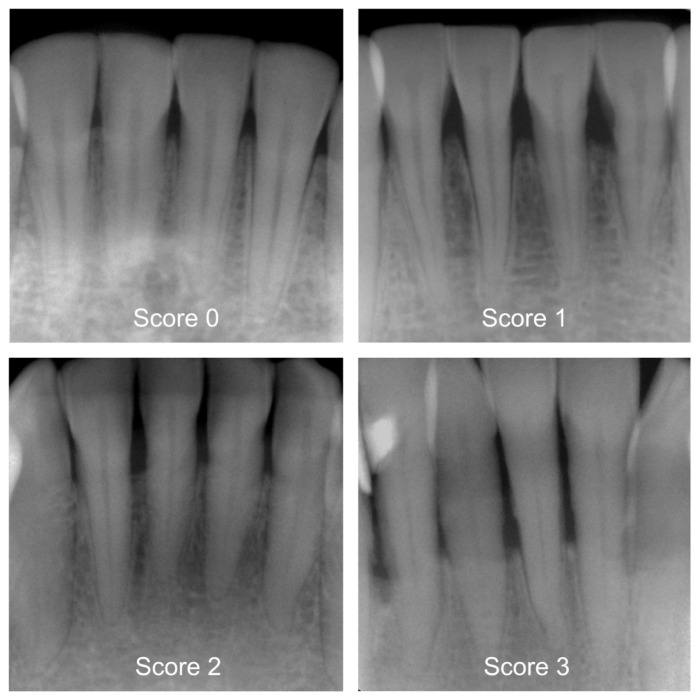
Examples of periapical radiographs for all categories: healthy periodontium, periodontal bone loss (PBL) not detectable (Score 0), mild radiographic PBL up to 15% in the coronal third of the tooth (Score 1), moderate radiographic PBL between 15% and 33% of the root length (Score 2), and severe radiographic PBL beyond the coronal third of the tooth (Score 3).

**Figure 2 jcm-12-07189-f002:**
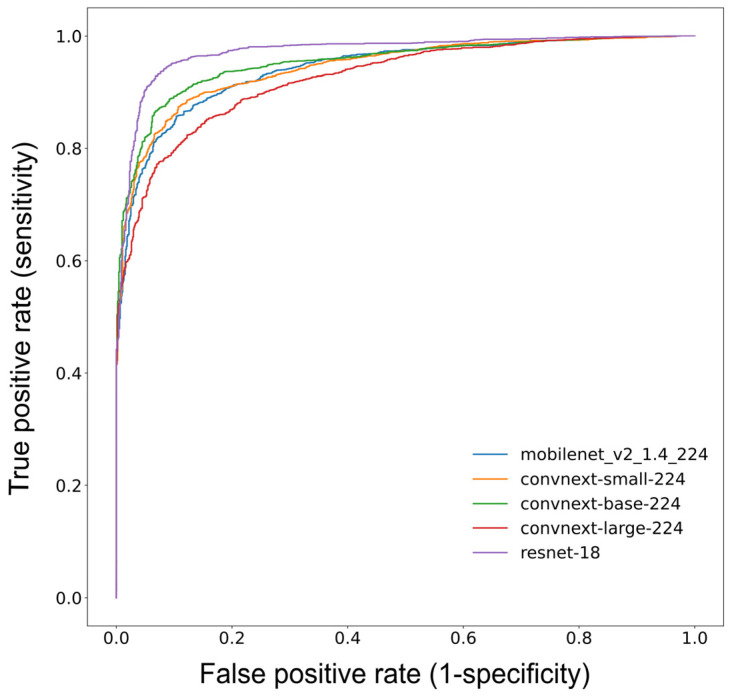
The receiver operating characteristic (ROC) curves graphically visualize the diagnostic performance of the developed convolutional neural networks (CNNs) in detecting PBL.

**Table 1 jcm-12-07189-t001:** Cohen’s kappa values for inter- and intra-examiner reliability for the detection of PBL, calculated among participating dentists (*N* = 7) in relation to the reference standard.

Examiner	Inter-Examiner	Intra-Examiner
P.H.	0.601–0.650	0.889
T.M.	0.620–0.658	0.554
A.W.	0.762–0.796	0.779
L.M.	0.516–0.565	0.797
U.W.	0.658–0.699	0.455
J.K.	0.706–0.748	0.579
H.D.	0.529–0.534	0.767

**Table 2 jcm-12-07189-t002:** Overview of the included periapical radiographs (*N* = 21,819) in relation to the corresponding sextants and periodontal diagnosis.

Expert Classification		Healthy Periodontium (Score 0)		Mild PBL (Score 1)		Moderate PBL (Score 2)		Severe PBL (Score 3)		Total	
		*N*	%	*N*	%	*N*	%	*N*	%	*N*	%
Upper jaw	Anteriors	653	3.0	661	3.0	433	2.0	197	0.9	1944	8.9
	1st Quadrant	1701	7.8	1826	8.4	851	3.9	367	1.7	4745	21.8
	2nd Quadrant	1231	5.6	2080	9.5	1093	5.0	312	1.5	4716	21.6
Lower jaw	Anteriors	202	0.9	676	3.1	786	3.6	325	1.5	1989	9.1
	3rd Quadrant	1477	6.8	2033	9.3	593	2.7	157	0.7	4260	19.5
	4th Quadrant	1282	5.9	2027	9.3	713	3.3	143	0.6	4165	19.1
	Total	6546	30.0	9303	42.6	4469	20.5	1501	6.9	21,819	100

**Table 3 jcm-12-07189-t003:** Overview of the true positive (TP), true negative (TN), false positive (FP), and false negative (FN) distribution for the independent test set (*N* = 3000 radiographs), which was evaluated by the AI-based algorithm for the assessment of periodontal bone loss.

CNN	True Positive (TP)		True Negative (TN)		False Positive (FP)		False Negative (FN)	
	*N*	%	*N*	%	*N*	%	*N*	%
ResNet-18	1876	62.5	609	20.3	294	9.8	221	7.4
MobileNetV2	1863	62.1	598	19.9	305	10.2	234	7.8
ConvNeXT/s ^1^	1877	62.6	639	21.3	264	8.8	220	7.3
ConvNeXT/b ^2^	1901	63.4	643	21.4	260	8.7	196	6.5
ConvNeXT/l ^3^	1890	63.0	637	21.2	266	8.9	207	6.9

^1^ small, ^2^ base, ^3^ large.

**Table 4 jcm-12-07189-t004:** Overview of the overall diagnostic performance of the developed convolutional neural network (CNN), where the independent test set (*N* = 3000 radiographs) was evaluated by the AI-based algorithm for the assessment of periodontal bone loss. The overall diagnostic accuracy (ACC), sensitivity (SE), specificity (SP), negative predictive value (NPV), positive predictive value (PPV), and area under the receiver operating characteristic curve (AUC) were predicted.

CNN	Diagnostic Performance					
	ACC	SE	SP	NPV	PPV	AUC
ResNet-18	82.8	89.5	67.4	73.4	86.5	0.884
MobileNetV2	82.0	88.8	66.2	71.9	85.9	0.884
ConvNeXT/s ^1^	83.9	89.5	70.8	74.4	87.7	0.903
ConvNeXT/b ^2^	84.8	90.7	71.2	76.6	88.0	0.911
ConvNeXT/l ^3^	84.2	90.1	70.5	75.5	87.7	0.913

^1^ small, ^2^ base, ^3^ large.

**Table 5 jcm-12-07189-t005:** Overview of the true positive (TP), true negative (TN), false positive (FP), and false negative (FN) distribution for the independent test set (*N* = 3000 radiographs) in different sextants, which was evaluated by the AI-based algorithm for the assessment of periodontal bone loss.

CNN	True Positive (TP)		True Negative (TN)		False Positive (FP)		False Negative (FN)	
	*N*	%	*N*	%	*N*	%	*N*	%
**Radiographs with maxillary anterior teeth**								
ResNet-18	155	58.7	72	27.3	27	10.2	10	3.8
MobileNetV2	154	58.3	79	29.9	20	7.6	11	4.2
ConvNeXT/s ^1^	155	58.7	79	29.9	20	7.6	10	3.8
ConvNeXT/b ^2^	157	59.5	77	29.2	22	8.3	8	3.0
ConvNeXT/l ^3^	158	59.8	74	28.0	25	9.5	7	2.7
**Radiographs with maxillary posterior teeth**								
ResNet-18	786	59.1	263	19.8	151	11.4	129	9.7
MobileNetV2	798	60.0	239	18.0	175	13.2	117	8.8
ConvNeXT/s ^1^	783	58.9	275	20.7	139	10.5	132	9.9
ConvNeXT/b ^2^	794	59.7	278	20.9	136	10.2	121	9.1
ConvNeXT/l ^3^	794	59.8	266	20.0	148	11.1	121	9.1
**Radiographs with mandibular anterior teeth**								
ResNet-18	244	89.7	14	5.2	11	4.0	3	1.1
MobileNetV2	239	87.9	19	7.0	6	2.2	8	2.9
ConvNeXT/s ^1^	242	89.0	19	7.0	6	2.2	5	1.8
ConvNeXT/b ^2^	244	89.7	17	6.3	8	2.9	3	1.1
ConvNeXT/l ^3^	243	89.3	18	6.6	7	2.6	4	1.5
**Radiographs with mandibular posterior teeth**								
ResNet-18	691	60.9	260	22.9	105	9.3	79	6.9
MobileNetV2	672	59.2	261	23.0	104	9.2	98	8.6
ConvNeXT/s ^1^	697	61.4	266	23.4	99	8.7	73	6.4
ConvNeXT/b ^2^	706	62.2	271	23.9	94	8.3	64	5.6
ConvNeXT/l ^3^	695	61.2	279	24.6	86	7.6	75	6.6

^1^ small, ^2^ base, ^3^ large.

**Table 6 jcm-12-07189-t006:** Overview of the diagnostic performance of the developed convolutional neural networks (CNNs) for different sextants, where the independent test set (*N* = 3000 radiographs) was evaluated by the AI-based algorithm for the assessment of periodontal bone loss. The overall diagnostic accuracy (ACC), sensitivity (SE), specificity (SP), negative predictive value (NPV), positive predictive value (PPV), and area under the receiver operating characteristic curve (AUC) were predicted.

	Diagnostic Performance					
	ACC	SE	SP	NPV	PPV	AUC
**Radiographs with maxillary anterior teeth**						
ResNet-18	86.0	93.9	72.7	87.8	85.2	0.925
MobileNetV2	88.3	93.3	79.8	87.8	88.5	0.935
ConvNeXT/s ^1^	88.6	93.9	79.8	88.8	88.6	0.951
ConvNeXT/b ^2^	88.6	95.2	77.8	90.6	87.7	0.959
ConvNeXT/l ^3^	87.9	95.8	74.7	91.4	86.3	0.950
**Radiographs with maxillary posterior teeth**						
ResNet-18	78.9	85.9	63.5	67.1	83.9	0.844
MobileNetV2	78.0	87.2	57.7	67.1	82.0	0.839
ConvNeXT/s ^1^	79.6	85.6	66.4	67.6	84.9	0.858
ConvNeXT/b ^2^	80.7	86.8	67.1	69.7	85.4	0.868
ConvNeXT/l ^3^	79.8	86.8	64.3	68.7	84.3	0.866
**Radiographs with mandibular anterior teeth**						
ResNet-18	94.9	98.8	56.0	82.4	95.7	0.942
MobileNetV2	94.9	96.8	76.0	70.4	97.6	0.960
ConvNeXT/s ^1^	96.0	98.0	76.0	79.2	97.6	0.969
ConvNeXT/b ^2^	96.0	98.8	68.0	85.0	96.8	0.978
ConvNeXT/l ^3^	96.0	98.4	72.0	81.8	97.2	0.980
**Radiographs with mandibular posterior teeth**						
ResNet-18	83.8	89.7	71.2	76.7	86.8	0.895
MobileNetV2	82.2	87.3	71.5	72.7	86.6	0.893
ConvNeXT/s ^1^	84.8	90.5	72.9	78.5	87.6	0.916
ConvNeXT/b ^2^	86.1	91.7	74.2	80.9	88.3	0.921
ConvNeXT/l ^3^	85.8	90.3	76.4	78.8	89.0	0.930

^1^ small, ^2^ base, ^3^ large.

## Data Availability

The data that support the findings of this study are available from the corresponding author upon reasonable request.

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
