# Peer review of "Detection of Periodontal Bone Loss on Periapical Radiographs—A Diagnostic Study Using Different Convolutional Neural Networks"

_jcm, 2023, doi:10.3390/jcm12227189_

Round 1

Reviewer 1 Report

Comments and Suggestions for Authors

Thanks to the editor for giving me the opportunity to review the manuscript. It is an interesting manuscript with a large sample size and does provide some new ideas to the readers. In this study, the diagnostic performance of five CNNs in detecting PBL on periapical radiographs was analysed. The results show that none of the five methods met the requirements (accuracy of at least 90%), suggesting that more efforts are needed for the clinical application of this method. This study has obvious reference value for the readers. However, there're some issues which should be addressed.

1. Why didn't the authors choose panoramic X-rays?  Please provide an explanation in the section of discussion.

2. The results showed that none of the five methods met the requirements (accuracy of at least 90%). Please provide possible causes, possible solutions, or future research directions in the section of discussion.

3. This is an X-ray only study, if the authors can increase the clinical manifestations in these patients (pocket formation, ulceration of the pocket epithelium, etc.) and a definitive diagnosis. Maybe more information could be provided.

4. There're too many data in the tables. It is suggested to change some tables into figures.

Author Response

Thank you for your feedback and helpful comments regarding our manuscript. We have carefully reviewed the comments and revised the manuscript accordingly. Please find our point-by-point responses in the attachment.

Reviewer 2 Report

Comments and Suggestions for Authors

“Comparison of different convolutional neural networks for the detection of periodontal bone loss on periapical radiographs” was submitted to JCM.

This study aimed to analyze the diagnostic performance of five CNNs in detecting PBL on periapical radiographs.

The authors concluded that automated assessment of PBL appears to be possible, but that diagnostic accuracy varies among different locations in the dentition.

The manuscript deals with an interesting issue; however, there are several concerns related to the study.

Title: Please include the type of study.

Abstract

Kindly provide the results for sensitivity, specificity, and AUC. Additionally, ensure that you do not introduce acronyms in the abstract that will not be utilized later.

Keywords: Please ensure that all of them correspond to MeSH terms.

Introduction

-Please incorporate the concept of dysbiosis into the definition of periodontitis.

-On lines 54-55, please specify the approach this study takes to address these issues.

-Regarding lines 59-62, please clarify whether you intend to formulate hypotheses.

Methods

-On line 78, please specify the number of participating dentists.

-In lines 103 and 115, provide a detailed description of the random selection process.

-On line 82, present the results of the Kappa test.

Line 84: Please describe if experienced examiners underwent a calibration process, and if so, include the concordance results.

-It is advisable to perform a sample size calculation.

Comments on the Quality of English Language

minor

Author Response

(The authors gave the same response as above.)

Reviewer 3 Report

Comments and Suggestions for Authors

Keywords should be addressed according to mesh terms and alphabetical order

Table 3 is so confusion and not readable shoule be transformed into many tables or should be spli into a diagramm

The study lacks of letter of significance for tavkes

Where is the sample size?

Please add more limitations and extend the introduction part too small and should be expanded

Comments on the Quality of English Language

Minor

Author Response

(The authors gave the same response as above.)
